# Deep Recurrent Neural Network-Based Identification of Precursor microRNAs

**Seunghyun Park**
Electrical and Computer Engineering
Seoul National University
Seoul 08826, Korea
School of Electrical Engineering
Korea University
Seoul 02841, Korea

**Seonwoo Min**
Electrical and Computer Engineering
Seoul National University
Seoul 08826, Korea

**Hyun-Soo Choi**
Electrical and Computer Engineering
Seoul National University
Seoul 08826, Korea

**Sungroh Yoon**∗
Electrical and Computer Engineering
Seoul National University
Seoul 08826, Korea
sryoon@snu.ac.kr

## Abstract

MicroRNAs (miRNAs) are small non-coding ribonucleic acids (RNAs) which play key roles in post-transcriptional gene regulation. Direct identification of mature miRNAs is infeasible due to their short lengths, and researchers instead aim at identifying precursor miRNAs (pre-miRNAs). Many of the known pre-miRNAs have distinctive stem-loop secondary structure, and structure-based filtering is usually the first step to predict the possibility of a given sequence being a pre-miRNA. To identify new pre-miRNAs that often have non-canonical structure, however, we need to consider additional features other than structure. To obtain such additional characteristics, existing computational methods rely on manual feature extraction, which inevitably limits the efficiency, robustness, and generalization of computational identification. To address the limitations of existing approaches, we propose a pre-miRNA identification method that incorporates (1) a deep recurrent neural network (RNN) for automated feature learning and classification, (2) multimodal architecture for seamless integration of prior knowledge (secondary structure), (3) an attention mechanism for improving long-term dependence modeling, and (4) an RNN-based class activation mapping for highlighting the learned representations that can contrast pre-miRNAs and non-pre-miRNAs. In our experiments with recent benchmarks, the proposed approach outperformed the compared state-of-the-art alternatives in terms of various performance metrics.

## 1 Introduction

MicroRNAs (miRNAs) play crucial roles in post-transcriptional gene regulation by binding to the $3'$ untranslated region of target messenger RNAs [16]. Among the research problems related to miRNA, computational identification of miRNAs has been one of the most significant. The biogenesis of a miRNA consists of the primary miRNA stage, the precursor miRNA (pre-miRNA) stage, and the mature miRNA stage [17]. Mature miRNAs are usually short, having only 20–23 base pairs (bp), and it is difficult to identify them directly. Most computational approaches focus on detecting

---

∗To whom correspondence should be addressed.

pre-miRNAs since they are usually more identifiable because they are longer (approximately 80bp) and have a distinctive stem-loop secondary structure.

In terms of the machine learning (ML), pre-miRNA identification can be viewed as a binary classification problem in which a given sequence must be classified as either a pre-miRNA or a non-pre-miRNA. A variety of computational approaches for miRNA identification have been proposed, and we can broadly classify them [18] into rule-based such as MIReNA [1], and ML-based approaches, which can be categorized into three groups in terms of the classification algorithm used: (1) MiPred [12], microPred [2], triplet-SVM [3], iMiRNA-SSF [38], miRNApre [39], and miRBoost [4] use support vector machines; (2) MiRANN [1] and DP-miRNA [37] use neural networks; and (3) (CSHMM) [5], which use a context-sensitive hidden Markov model.

Known pre-miRNAs have distinctive structural characteristics, and therefore most computational methods make first-order decisions based on the secondary structure of the input RNA sequence. However, the identification of new pre-miRNAs with non-canonical structure and subtle properties, and maybe both, it requires the consideration of features other than secondary structure. Some authors [19] have even argued that the performance of ML-based tools are more dependent on the set of input features than the ML algorithms that are used.

The discovery of new features which are effective in pre-miRNA identification currently involves either searching for hand-crafted features (such as the frequency of nucleotide triplets in the loop, global and intrinsic folding attributes, stem length, and minimum free energy) or combining existing features. One recent study utilized 187 such features [4], another 48 features [2], most of which were manually prepared. Manual feature extraction requires ingenuity and inevitably limits the efficiency, robustness, and generalization of the resulting identification scheme developed. Furthermore, neural network-based methods above only use neural networks for classification of hand-designed features, and not for feature learning.

Similar challenges exist in other disciplines. Recently, end-to-end deep learning approaches have been successfully applied to tasks such as speech and image recognition, largely eliminating the manual construction of feature engineerings. Motivated by these successes, we propose a deep neural network-based pre-miRNA identification method which we call deepMiRGene to address the limitations of existing approaches. It incorporates the following key components:

1. A deep recurrent neural network (RNN) with long short-term memory (LSTM) units for RNA sequence modeling, automated feature learning, and robust classification based on the learned representations.
2. A multimodal architecture for seamless integration of prior knowledge (such as the importance of RNA secondary structure in pre-miRNA identification) with automatically learned features.
3. An attention mechanism for effective modeling of the long-term dependence of the primary structure (i.e., sequence) and the secondary structure of RNA molecules.
4. An RNN-based class activation mapping (CAM) to highlight the learned representations in the way that contrasts pre-miRNAs and non-pre-miRNAs to obtain biological insight.

We found that simply combining existing deep learning modules did not deliver satisfactory performance in our task. Thus our contribution can be seen as inventing a novel pipeline and with components optimized for handling RNA sequences and structures to predict (possibly subtle) pre-miRNA signals, rather than just assembling pre-packaged components. Our research for an optimal set of RNN architectures and hyperparameters for pre-miRNA identification involved an exploration of the design space spanned by the components of our methodology. The result of this research is a technique with demonstrable advantages over other state-of-the-art alternatives in terms of both cross-validation results but also the generalization ability (i.e., performance on test data). The source code for the proposed method is available at https://github.com/eleventh83/deepMiRGene.

## 2 Related Work

### 2.1 The Secondary Structure of a Pre-miRNA

The secondary structure of an RNA transcript represents the base-pairing interactions within that transcript. The usual secondary structure of a pre-miRNA is shown in Fig. 1, which shows that a pre-miRNA is a base-paired double helix rather than a single strand, and this pairing is one

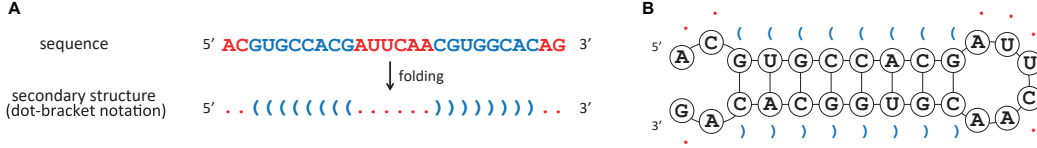

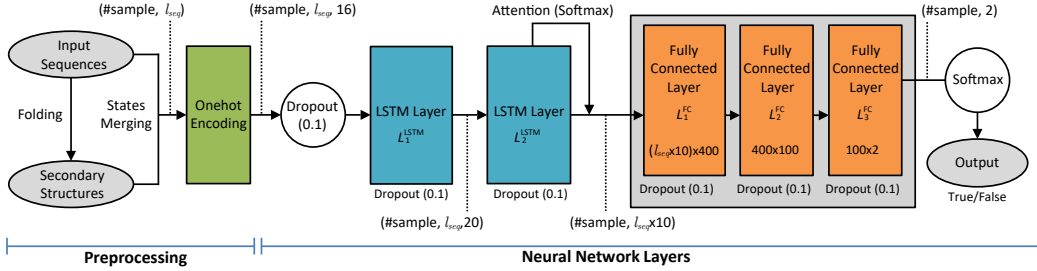

Figure 1: (A) sequence of a pre-miRNA, and (B) the secondary structure of the given sequence. The dot-bracket notation in (A) describes RNA secondary structures. Unpaired nucleotides are represented as "."s and base-paired nucleotides are represented as "("s and ")"s.

Figure 2: Overview of our method: #sample is the number of input sequences and $l_{seq}$ is the maximum length of the input sequence. The dimension of intermediate data is labeled (#sample, $l_{seq}$, 16).

of the most prominent features for pre-miRNA identification [12, 2]. The secondary structure of a given sequence can be predicted by tools such as RNAfold [5], which is widely used. It constructs a thermodynamically stable secondary structure from a given RNA sequence by calculating the minimum free energy and the probable base-pairings [20]. However, reliable pre-miRNA identification requires features other than the secondary structure to be considered, since false positives may be generated due to the limitations of structure prediction algorithms and the inherent unpredictability of these structures [21].

## 2.2 Deep Recurrent Neural Networks

RNNs are frequently used for sequential modeling and learning. RNNs process one element of input data at a time and implicitly store past information using cyclic connections of hidden units [8]. However, early RNNs often had difficulty in learning long-term dependencies because of the vanishing or exploding gradient problem [9]. Recent deep RNNs incorporate mechanisms to address this problem. Explicit memory units, such as LSTM units [10] or GRUs [3], are one such mechanism. An LSTM unit, for works as a sophisticated hidden unit that uses multiplicative gates to learn when to input, output, and forget in addition to cyclic connections to store the state vector. A more recent innovation [2] is the attention mechanism. This can take various forms, but in our system, a weighted combination of the output vectors at each point in time replaces the single final output vector of a standard RNN. An attention mechanism of this sort helps learn long-term dependencies and also facilitates the interpretation of results, e.g., by showing how closely the output at a specific time point is related to the final output [30, 29, 2].

## 3 Methodology

Fig. 2 shows the proposed methodology of our system. The input consists of either a set of pre-miRNA sequences (in the training phase) or a test sequence (in the testing phase). The output for each input sequence is a two-dimensional (softmax) vector which indicates whether the input sequence encodes pre-miRNA or not. In a preprocessing phase, we derive the secondary structure of the input sequence and then encode the sequence and its structure together into a 16-dimensional binary vector. Encoded vectors are then processed by the RNN architecture, consisting of LSTM layers and fully connected (FC) layers, and the attention mechanism. The pseudocode of our approach is available as Appendix A, in the supplementary material.

## 3.1 Preprocessing

Preprocessing a set of input pre-miRNA sequences involves two tasks. First, RNAfold is used to obtain the secondary structure of each sequence; we already noted the importance of this data. We clarify that each position in an RNA sequence as one of $\{$A, C, G, U$\}$, and the corresponding location in the secondary structure as one of $\{$(, ), ., :$\}$. This dot-bracket notation is shown in Fig. 1. The symbol ":" represents a position inside a loop (unpaired nucleotides surrounded by a stem). Let $x_s$ and $x_t$ denote an input sequence and its secondary structure, respectively. Then, $x_s \in \{$A, C, G, U$\}^{|x_s|}$ and $x_t \in \{$(, ), ., :$\}^{|x_t|}$. Note that $|x_s| = |x_t|$.

Next, each input sequence $x_s$ is combined with its secondary structure $x_t$ into a numerical representation. This is a simple one-hot encoding [4], which gave better results in our experiments than a soft encoding (see Section 4). Our encoding scheme uses a 16-dimensional one-hot vector, in which position $i$ ($i = 0, 1, \ldots, 15$) is interpreted as follows:

$$
\text{if } \lfloor i/4 \rfloor = \begin{cases} 0 & \text{then A} \\ 1 & \text{then G} \\ 2 & \text{then C} \\ 3 & \text{then U} \end{cases} \text{ and if } i\%4 = \begin{cases} 0 & \text{then (} \\ 1 & \text{then )} \\ 2 & \text{then .} \\ 3 & \text{then :} \end{cases}
$$

The "%" symbol denotes the modulus operator. After preprocessing, the sequence $x_s$ and the structure $x_t$ are together represented by the matrix $X_s \in \{0, 1\}^{|x_s| \times 16}$, each row of which is the 16-dimensional one-hot vector described above. For instance, $x_s = $ AUG and $x_t = $ (:) are represented by the following $3 \times 16$ binary matrix:

$$
X_s = \begin{bmatrix} \overbrace{1\,0\,0\,0}^{\text{A}} & \overbrace{0\,0\,0\,0}^{\text{C}} & \overbrace{0\,0\,0\,0}^{\text{G}} & \overbrace{0\,0\,0\,0}^{\text{U}} \\ 0\,0\,0\,0 & 0\,0\,0\,0 & 0\,0\,0\,0 & 0\,0\,0\,1 \\ \underbrace{0\,0\,0\,0}_{\text{()., :}} & 0\,0\,0\,0 & 0\,1\,0\,0 & 0\,0\,0\,0 \end{bmatrix}.
$$

## 3.2 Neural Network Architecture

The main features of our neural network is the attention mechanism provided by the LTSM and FC layers.

**1) LSTM layers:** The purpose of these layers is sequential modeling of the primary and secondary structure of the input pre-miRNA transcripts. We use two stacked LSTM layers denoted by $L_1^{\text{LSTM}}$ and $L_2^{\text{LSTM}}$ respectively. $L_1^{\text{LSTM}}$ takes the matrix $X_s$ produced in the preprocessing stage and returns a weight matrix $H_1$, as follows:

$$
H_1 = L_1^{\text{LSTM}}(X_s) \in \mathbb{R}^{|x_s| \times d_1}, \tag{1}
$$

where $d_1$ is the number of LSTM units in the first layer. Similarly, the second layer first returns a second weight matrix $H_2$:

$$
H_2 = L_2^{\text{LSTM}}(H_1) \in \mathbb{R}^{|x_s| \times d_2}, \tag{2}
$$

where $d_2$ is the number of LSTM units in the second layer.

We apply an attention mechanism to the output of $L_2^{\text{LSTM}}$ with the aim of learning the importance of each position of $x_s$. The neural networks first learn an attention weight for each output of the second LSTM layer for each sequence position in a training process. These weights are collectively represented by a matrix $\Omega \in \mathbb{R}^{d_2 \times |x_s|}$. An attention weight matrix $\Omega_{\text{att}} \in \mathbb{R}^{|x_s| \times |x_s|}$ is then constructed as follows:

$$
\Omega_{\text{att}} = H_2 \Omega. \tag{3}
$$

This yields the attention weight vector $\omega_{\text{att}}$

$$
\omega_{\text{att}} = \text{softmax}(\text{diag}(\Omega_{\text{att}})) \in \mathbb{R}^{|x_s|}, \tag{4}
$$

where the $i^{\text{th}}$ element of $\omega_{\text{att}}$ corresponds to the attention weight for the $i^{\text{th}}$ position of $x_{\text{s}}$. Then, $H_{\text{att}} \in \mathbb{R}^{|x_{\text{s}}| \times d_2}$, the attention-weighted representation of $H_2$, can be expressed as follows:

$$H_{\text{att}} = H_2 \odot (\omega_{\text{att}} \otimes u_{d_2}),\qquad(5)$$

where $u_{d_2}$ is the $d_2$-dimensional unit vector, and $\odot$ and $\otimes$ respectively denote the element-wise multiplication and outer product operators.

Finally, we reshape the matrix $H_{\text{att}}$ by flattening it into a $(d_2 \cdot |x_{\text{s}}|)$-dimensional vector $\tilde{h}_{\text{att}}$ for the sake of compatibility with third-party software. We use the standard nonlinearities (i.e., hyperbolic tangent and logistic sigmoid) inside each LSTM cell.

**2) Fully connected layers:** The neural network collects the outputs from the last LSTM layer and makes a final decision using three FC layers. We denote the operations performed by these three layers by $L_1^{\text{FC}}$, $L_2^{\text{FC}}$, and $L_3^{\text{FC}}$, which allows us to represent the outputs of the three FC layers as $f_1 = L_1^{\text{FC}}\left(\tilde{h}_{\text{att}}\right)$, $f_2 = L_2^{\text{FC}}(f_1)$, and $\hat{y} = L_3^{\text{FC}}(f_2)$, where $f_1 \in \mathbb{R}^{d_3}$ and $f_2 \in \mathbb{R}^{d_4}$ are intermediate vectors, and $\hat{y} \in \mathbb{R}^2$ denotes the final softmax output; $d_3$ and $d_4$ are the numbers of hidden nodes in the last two FC layers. The first two FC layers use logistic sigmoids as theirs activation functions, while the last FC layer uses the softmax function.

### 3.3 Training

We based our training objective on binary cross-entropy (also known as logloss). As will be explained in Section 4 (see Table 1), we encountered a class-imbalance problem in this study, since there exist significantly more negative training examples (non-pre-miRNA sequences) than positives (known pre-miRNA sequences). We addressed this issue by augmenting the logloss training objective with balanced class weights [31], so that the training error $E$ is expressed as follows:

$$E = -\frac{1}{b} \sum_i \left\{ c^- y_i \log(\hat{y}_i) + c^+ (1 - y_i) \log(1 - \hat{y}_i) \right\}$$

where $b$ is the mini-batch size (we used $b = 128$), and $y_i \in \{0, 1\}$ is the class label provided in training data ($y_{\text{i}} = 0$ for pre-miRNA; $y_{\text{i}} = 1$ for non-pre-miRNA); $c^-$ and $c^+$ represent the balanced class weights given by

$$c^k = \frac{N}{2 \cdot n_k}, \qquad\qquad k \in \{-, +\} \qquad(6)$$

where $N$ is the total number of training examples and $n_k$ is the number of examples in either the positive or the negative class.

We minimized $E$ using the Adam [6] gradient descent method, which uses learning rates which adapt to the first and second moments of the gradients of each parameter. We tried other optimization methods (e.g., the stochastic gradient descent [27] and RMSprop [28]), but they did not give better results.

We used dropout regularization with an empirical setup. In the LSTM layers, a dropout parameter for input gates and another for recurrent connection were both set to $0.1$. In the FC layers, we set the dropout parameter to $0.1$. We tried batch normalization [22], but did not find it effective.

All the weights were randomly initialized in the range of $[-0.05, 0.05]$. The number of hidden nodes in the LSTM ($d_1$, $d_2$) and the FC ($d_3$, $d_4$) layers were determined by cross validation as $d_1 = 20$, $d_2 = 10$, $d_3 = 400$, and $d_4 = 100$. The mini-batch size and training epochs were set to 128 and 300 respectively.

## 4 Experimental Results

We used three public benchmark datasets [4] named *human*, *cross-species*, and *new*. The positive pre-miRNA sequences in all three datasets were obtained from miRBase [25] (release 18). For the negative training sets, we obtained noncoding RNAs other than pre-miRNAs and exonic regions of protein-coding genes from NCBI (http://www.ncbi.nlm.nih.gov), fRNAdb [23], NONCODE [24], and

Table 1: Numbers of sequences in the three benchmark datasets [4] used in this study. The median length of each dataset is given in brackets.

| Type \ Dataset name | *Human* | *Cross-species* | *New* |
|---|---|---|---|
| Positive examples | 863 (85) | 1677 (93) | 690 (71) |
| Negative examples | 7422 (92) | 8266 (96) | 8246 (96) |

Table 2: Performance evaluation of different pre-miRNA identification methods with cross-validation (CV) and test data using sensitivity (SE), specificity (SP), positive predictive value (PPV), F-score, geometric mean (g-mean), area under the receiver operating characteristic curve (AUROC), and area under the precision-recall curve (AUPR).

| | *Human* | | | | | | | *Cross-species* | | | | | | |
|---|---|---|---|---|---|---|---|---|---|---|---|---|---|---|
| Methods | SE[1] | SP[2] | PPV[3] | F-score[4] | g-mean[5] | AUROC | AUPR | SE | SP | PPV | F-score | g-mean | AUROC | AUPR |
| miRBoost (CV) | 0.803 | 0.988 | 0.887 | **0.843** | **0.891** | - | - | 0.861 | 0.977 | 0.884 | 0.872 | 0.917 | - | - |
| CSHMM (CV) | 0.713 | 0.777 | 0.559 | 0.570 | 0.673 | - | - | 0.826 | 0.576 | 0.533 | 0.564 | 0.524 | - | - |
| triplet-SVM (CV) | 0.669 | 0.986 | 0.851 | 0.749 | 0.812 | 0.957 | 0.854 | 0.735 | 0.967 | 0.819 | 0.775 | 0.843 | 0.943 | 0.869 |
| microPred (CV) | 0.763 | **0.989** | **0.888** | 0.820 | 0.869 | 0.974 | 0.890 | 0.825 | 0.975 | 0.875 | 0.848 | 0.897 | 0.970 | 0.873 |
| MIReNA (CV) | **0.818** | 0.943 | 0.624 | 0.708 | 0.878 | - | - | 0.766 | 0.952 | 0.765 | 0.765 | 0.854 | - | - |
| Proposed (CV) | 0.799 | 0.988 | 0.885 | 0.839 | 0.888 | **0.984** | **0.915** | **0.886** | **0.982** | **0.911** | **0.898** | **0.933** | **0.985** | **0.927** |
| miRBoost (test) | **0.884** | 0.969 | 0.768 | 0.822 | **0.925** | - | - | 0.856 | 0.844 | 0.526 | 0.651 | 0.850 | - | - |
| CSHMM (test) | 0.616 | 0.978 | 0.768 | 0.684 | 0.777 | - | - | 0.749 | 0.960 | 0.791 | 0.769 | 0.848 | - | - |
| triplet-SVM (test) | 0.744 | **0.992** | 0.914 | 0.821 | 0.859 | 0.947 | 0.830 | 0.760 | 0.977 | 0.870 | 0.812 | 0.862 | 0.952 | 0.908 |
| microPred (test) | 0.779 | 0.988 | 0.882 | 0.827 | 0.877 | 0.980 | 0.892 | 0.814 | **0.985** | **0.919** | 0.863 | 0.896 | 0.963 | 0.906 |
| MIReNA (test) | 0.826 | 0.941 | 0.617 | 0.706 | 0.881 | - | - | 0.796 | 0.950 | 0.764 | 0.780 | 0.870 | - | - |
| Proposed (test) | 0.822 | **0.992** | **0.919** | **0.868** | 0.903 | **0.981** | **0.918** | **0.900** | 0.983 | 0.913 | **0.906** | **0.940** | **0.984** | **0.955** |

TP: $\sum$ true positive, TN: $\sum$ true negative, FP: $\sum$ false positive, FN: $\sum$ false negative. [1] SE $=$ TP/(TP + FN) [2] SP $=$ TN/(TN + FP) [3] PPV (precision) $=$ TP/(TP + FP) [4] F-score $=$ 2TP/(2TP + FP + FN) [5] g-mean $= \sqrt{\text{SE} \cdot \text{SP}}$

snoRNA-LBME-db [26]. Note that we only acquired those datasets that had undergone redundancy removal and had annotation corrected by the data owners.

As shown in Table 1, the *human* dataset contains 863 human pre-miRNA sequences (positive examples) and 7422 non-pre-miRNA sequences (negative examples). The *cross-species* dataset contains 1677 pre-miRNA sequences collected from various species (e.g., human, mouse, and fly), and 8266 non-miRNA sequences. The *new* dataset has 690 newly discovered pre-miRNA sequences, which are in miRBase releases 19 and 20, with 8246 non-pre-miRNA sequences. For the *human* and *cross-species* datasets, 10% of the data was randomly chosen as a clean test dataset (also known as a publication dataset) and was never used in training. Using the remaining 90% of each dataset, we carried out five-fold cross-validation for training and model selection. Note that the *new* dataset was used for testing purposes only, as described in Tran et al. [4]. Additional details of the experimental settings used can be found in Appendix B.

## 4.1 Validation and Test Performance Evaluation

We used seven evaluation metrics: sensitivity (SE), specificity (SP), positive predictive value (PPV), F-score, the geometric mean of SE and SP (g-mean), the area under the receiver operating characteristic curve (AUROC), and the area under the precision-recall curve (AUPR). Higher sensitivity indicates a more accurate pre-miRNAs predictor which is likely to assist the discovery of novel pre-miRNAs. Higher specificity indicates more effective filtering of pseudo pre-miRNAs, which increases the efficiency of biological experiments. Because they take account of results with different decision thresholds, AUROC and AUPR typically deliver more information than the more basic metrics such as sensitivity, specificity, and PPV, which are computed with a single decision threshold. Note that miRBoost, MIReNA, and CSHMM do not provide decision values, and so the AUROC and AUPR metrics cannot be obtained from these methods.

The results of a **cross-validation performance** comparison are shown in the upper half of Table 2, while the results of the **test performance** comparison are shown in the bottom half. For the *human* dataset, the cross-validation performance of our method was comparable to that of others, but our method achieved the highest test performance in terms of F-score, AUROC, and AUPRG. For the *cross-species* dataset, our method achieved the best overall performance in terms of both cross-validation and test evaluation results. Some tools, such as miRBoost, showed fair performance in

Table 3: Evaluation of performance on the *new* dataset.

| Methods | SE | SP | PPV | F-score | g-mean | AUROC | AUPR |
|---|---|---|---|---|---|---|---|
| miRBoost | **0.921** | 0.936 | 0.609 | 0.733 | 0.928 | - | - |
| CSHMM | 0.536 | 0.069 | 0.046 | 0.085 | 0.192 | - | - |
| triplet-SVM | 0.721 | **0.981** | **0.759** | 0.740 | 0.841 | 0.934 | 0.766 |
| microPred | 0.728 | 0.970 | 0.672 | 0.699 | 0.840 | 0.940 | 0.756 |
| MIReNA | 0.450 | 0.941 | 0.392 | 0.419 | 0.650 | - | - |
| Proposed method | 0.917 | 0.964 | 0.682 | **0.782** | **0.941** | **0.981** | **0.808** |

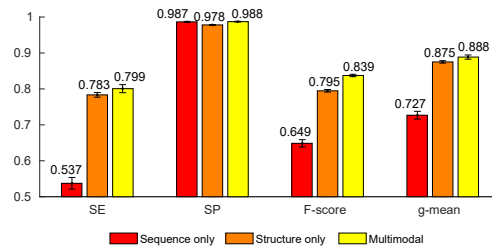

Figure 3: Using both sequence and structure information gives the best performance on the *human* dataset. Each bar shows the metrics of average cross-validation results.

terms of the cross-validation but failed to deliver the same level of performance on the test data. These results suggest that our approach provides better generalization than the alternatives. The similarity of the performance in terms of the cross-validation and test results suggests that overfitting was handled effectively.

Following the experimental setup used by Tran et al. [4], we also evaluated the proposed method on the *new* dataset, with a model trained by the *cross-species* dataset, as shown in Table 3, to assess the potential of our approach in the search for novel pre-miRNAs. Again, our method did not show the best performance in terms of basic metrics such as sensitivity and specificity, but it returned the best values of AUROC and AUPR. The results show that the proposed method can be used effectively to identify novel pre-miRNAs as well as to filter out pseudo pre-miRNAs.

To evaluate the statistical significance of our approach, we applied a Kolmogorov-Smirnov test [40] to the classifications produced by our method, grouped by true data labels. For the *human*, *cross-species*, and *new* datasets, the $p$-values we obtained were $5.23 \times 10^{-54}$, $6.06 \times 10^{-102}$, and $7.92 \times 10^{-49}$ respectively, indicating that the chance of these results occurring at random is very small indeed.

## 4.2   Effectiveness of Multimodal Learning

Our approach to the identification of pre-miRNAs takes both biological sequence information and secondary structure information into account. To assess the benefit of this multimodality, we measured the performance of our method using only sequences or secondary structures in training on the *human* dataset. As shown in Fig. 3, all of the performance metrics were higher when both sequence and structure information were used together. Compared with the use of sequence or structure alone, the sensitivity of the multimodal approach was increased by 48% point and 2% point, respectively. For specificity, the cases using both sequence and structure achieved higher performance values (0.988) than those of the sequence only (0.987) and structure only (0.978) cases. Similarly, in terms of F-score, using the multimodality gave 29% point and 5% point higher scores (0.839) than using sequence only (0.649) or structure only (0.795), respectively.

## 4.3   Gaining Insights by Analyzing Attention Weights

A key strength of our approach is its ability to learn the features useful for pre-miRNA identification from data. This improves efficiency, and also has the potential to aid the discovery of subtle features that might be missed in manual feature design. However, learned features, which are implicitly represented by the trained weights of a deep model, come without intuitive significance.

To address this issue, we experimented with the visualization of attention weights using the class activation mapping [32], a technique that was originally proposed to interpret the operation of

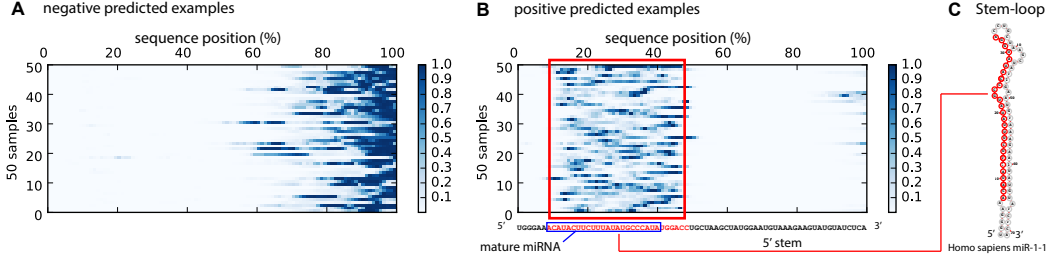

Figure 4: Attention weighted RNN outputs with the *human* dataset. (A) Class activation mapping for predicted examples (negatives: non-pre-miRNAs). (B) Class activation mapping for predicted examples (positives: pre-miRNAs). (C) Stem-loop structure of a pre-miRNA (*homo sapiens* miR-1-1).

Table 4: Performance of different types of neural network, assessed in terms of five-fold cross-validation results from the *human* dataset. The number of stacked layers is shown in brackets. ATT means that an attention mechanism was included, and a BiLSTM is a bi-directional LSTM. The configuration that we finally adopted is shown in row 6.

| No. | Type | SE | SP | F-score | g-mean |
|-----|------|-----|-----|---------|--------|
| 1 | 1D-CNN(2) | 0.745 | 0.978 | 0.771 | 0.853 |
| 2 | 1D-CNN(2)+LSTM(2) | 0.707 | 0.976 | 0.738 | 0.830 |
| 3 | 1D-CNN(2)+LSTM(2)+ ATT | 0.691 | 0.979 | 0.739 | 0.822 |
| 4 | LSTM(2) | 0.666 | **0.988** | 0.751 | 0.810 |
| 5 | LSTM(1) + ATT | 0.781 | 0.987 | 0.824 | 0.878 |
| 6 | LSTM(2) + ATT (proposed) | **0.799** | **0.988** | **0.839** | **0.888** |
| 7 | BiLSTM(1) + ATT | 0.783 | 0.987 | 0.827 | 0.879 |
| 8 | BiLSTM(2) + ATT | 0.795 | 0.987 | 0.834 | 0.886 |

convolutional neural networks (CNNs) in image classification by highlighting discriminative regions. We modified the class activation mapping of RNNs to discover which part of the sequential output is significant for identifying pre-miRNAs. We performed one-dimensional global average pooling (GAP) on the attention weighted output $H_{att}$ (see Section 3.2) to derive a $d_2$-dimensional weight vector $\omega_{gap}$. We then multiplied $H_{att}$ by $\omega_{gap}$ to obtain a class activation map of size $|x_s|$ for each sequence sample.

Fig. 4 (A) and Fig. 4 (B) show the resulting heatmap representations of class activation mapping on the *human* dataset for positive and negative predicted examples, respectively. Since sequences can have different lengths, we normalized the sequence lengths to 1 and presented individual positions in a sequence between 0% and 100% in the $x$-axis. By comparing the plots in Fig. 4 (A) and (B), we can see that class activation maps of the positive and negative data show clear differences, especially at the 10–50% sequence positions, within the red box in Fig. 4 (B). This region corresponds to the $5'$ stem region of typical pre-miRNAs, as shown in Fig. 4 (C). This region coincides with the location of a mature miRNA encoded within a pre-miRNA, suggesting that the data-driven features learned by our approach have revealed relevant characteristics of pre-miRNAs.

The presence of some nucleotide patterns has recently been reported in the mature miRNA region inside a pre-miRNA [33]. We anticipate that further interpretation of our data-driven features may assist in confirming such patterns, and also in discovering novel motifs in pre-miRNAs.

### 4.4 Additional Experiments

**1) Architecture exploration:** We explored various alternative network architectures, as listed in Table 4, which shows the performance of different network architectures, annotated with the number of layers and any of attention mechanism. Rows 1–3 of the table, show results for CNNs with and without LSTM networks. Rows 4–6 show the results of LSTM networks. Rows 7–8, show results for bi-directional LSTM (BiLSTM) networks. More details can be found in Appendix C.1.

**2) Additional results:** Appendix C.2–4 presents more details of hyperparameter tuning, the design decisions made between the uses of soft and hard encoding, and running-time comparisons.

# 5 Discussion

Given the importance of the secondary structure in pre-miRNA identification (e.g., see Section 4.2), we derived the secondary structure of each input sequence using RNAfold. We then combined the secondary structure information with the primary structure (i.e. the sequence), and sent the result to the RNN. However, a fully end-to-end approach to pre-miRNA identification we would need to learn even the secondary structure from the input sequences. Due to the limited numbers of known pre-miRNA sequences, this remains as challenging future work.

Our experimental results supported the effectiveness of a multi-modal approach that considers sequences and structures together from an early stage of the pipeline. Incorporating other types of information would be possible and might improve performance further. For example, sequencing results from RNA-seq experiments reflect the expression levels and the positions of each sequenced RNA [34]; and conservation information would allow a phylogenetic perspective [35]. Such additional information could be integrated into the current framework by representing it as new network branches and merging them with the current data before the FC layers.

Our proposed method has the clear advantage over existing approaches that it does not require hand-crafted features. But we need to ensure that learned feature provide satisfactory performance, and they also need to have some biological meaning. Biomedical researchers naturally hesitate to use a black-box methodology. Our method of visualizing attention weights provides a tool for opening that black-box, and assist data-driven discovery.

**Acknowledgments**

This work was supported in part by the Samsung Research Funding Center of the Samsung Electronics [No. SRFC-IT1601-05], the Institute for Information & communications Technology Promotion (IITP) grant funded by the Korea government (MSIT) [No. 2016-0-00087], the Future Flagship Program funded by the Ministry of Trade, Industry & Energy (MOTIE, Korea) [No. 10053249], the Basic Science Research Program through the National Research Foundation of Korea (NRF) funded by the Ministry of Science, ICT & Future Planning [No. 2016M3A7B4911115], and Brain Korea 21 Plus Project in 2017.

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
