[Supplementary Material]

# Appendix

## A  Pseudocode of the Proposed Algorithm

---

**Algorithm 1** Pseudocode of our proposed methodology for pre-miRNA prediction

---

1: **Input:** $x_s \in \{A, C, G, U\}^{|x_s|}$, $y \in \{[0\ 1]^\top, [1\ 0]^\top\}$
   ▷ $x_s$: a pre-miRNA sequence with its length of $|x_s|$.
   ▷ $y$: A 2-dim vector indicating the true label for $x_s$.
2: **Param:** $b$: mini-batch size
3: **Output:** whole model weights $W$ and prediction $\hat{y} \in \mathbb{R}^2$. $W$ is composed of the weights of the two LSTM layers, attention mechanism, and three fully connected layers.

   **Step 1: Preprocessing (Section 3.1)**
4: **for** each sequence $x_s$ in training samples
5:     $x_t \leftarrow \text{fold}(x_s)$
       ▷ fold: a function to predict the secondary structure of a given input.
       ▷ $x_t$: secondary structure of given input sequence $x_s$ in dot-bracket notation. $|x_t| = |x_s|$ and $x_t \in \{(, ), ., :\}^{|x_t|}$.
6:     $X_s \leftarrow \text{encode}(x_s, x_t)$
       ▷ encode: a function to convert a given sequence and structure into a 16-dimensional one-hot encoded matrix.
       ▷ $X_s$: a one-hot encoded matrix. $X_s \in \{0, 1\}^{|x_s| \times 16}$.

   **Step 2: Training on neural network (Section 3.2)**
7: initialize weights $W$
8: **for** each epoch
9:     **for** $b$ training data composed of $X_s$, which is randomly picked from training samples

       **LSTM Layers with Attention Mechanism**
10:        $H_1 \leftarrow L_1^{\text{LSTM}}(X_s)$
           ▷ $H_1 \in \mathbb{R}^{|x_s| \times d_1}$, where $d_1$ is the no. of LSTM units in the 1st layer.
11:        $H_2 \leftarrow L_2^{\text{LSTM}}(H_1)$
           ▷ $H_2 \in \mathbb{R}^{|x_s| \times d_2}$, where $d_2$ is the no. of LSTM units in the 2nd layer.
12:        $\Omega_{\text{att}} \leftarrow H_2 \Omega$

           ▷ $\Omega \in \mathbb{R}^{d_2 \times |x_s|}$: learned weights for the output of the second LSTM layer; $\Omega_{\text{att}} \in \mathbb{R}^{|x_s| \times |x_s|}$: attention weight matrix.
13:        $\omega_{\text{att}} \leftarrow \text{softmax}(\text{diag}(\Omega_{\text{att}}))$
           ▷ $\omega_{\text{att}} \in \mathbb{R}^{|x_s|}$: attention weights for the individual positions in $x_s$.
14:        $H_{\text{att}} \leftarrow H_2 \odot (\omega_{\text{att}} \otimes u_{d_2})$
           ▷ $H_{\text{att}} \in \mathbb{R}^{|x_s| \times d_2}$: attention-weighted representation of $H_2$.
           ▷ $\odot$: element-wise multiplication.
           ▷ $\otimes$: outer product operator.
15:        $\tilde{h}_{\text{att}} \leftarrow \text{flatten}(H_{\text{att}})$
           ▷ $\tilde{h}_{\text{att}} \in \mathbb{R}^{d_2 \cdot |\tilde{x}_s|}$: final output vector of the LSTM layers.
           ▷ flatten: reshape a given matrix into a vector.

           **Fully Connected Layers**
16:        $f_1 \leftarrow L_1^{\text{FC}}(\tilde{h}_{\text{att}})$
           ▷ $f_1 \in \mathbb{R}^{d_3}$ (sigmoid activated).
17:        $f_2 \leftarrow L_2^{\text{FC}}(f_1)$
           ▷ $f_2 \in \mathbb{R}^{d_4}$ (sigmoid activated).
18:        $\hat{y} \leftarrow L_3^{\text{FC}}(f_2)$
           ▷ $\hat{y} \in \mathbb{R}^2$ (softmax activated).
           ▷ $\hat{y}$: final output.

           **Weight Update (Section 3.3)**
19:        $E \leftarrow -\frac{1}{b} \sum_i \{c^- y_i \log(\hat{y}_i) + c^+ (1-y_i) \log(1-\hat{y}_i)\}$
           ▷ $E$: mini-batch training error obtained using binary cross-entropy (logloss).
           ▷ $b$: mini-batch size ($b = 128$).
           ▷ $c^-$, $c^+$: class weights.
20:        $W \leftarrow W - \triangle W$
           ▷ calculate $\triangle W$ based on $E$ using gradient descent optimization algorithm "Adam"

---

## B  Experimental Setup

Our method was implemented using Theano [6, 7] and Keras [8]. All the experiments were performed on a system equipped with an Intel Xeon E5-2650 CPU, 8 GB of memory, and an Nvidia Geforce Titan X GPU with 12 GB of memory and 3072 cores. For comparison with our method, we examined the performance of five state-of-the-art tools: miRBoost [4], CSHMM [5], triplet-SVM [3], microPred [2], and MIReNA [1]. The last is a rule-based tool supporting pre-determined thresholds for prediction; allowing us to skip training and perform inference directly with the validation data. The other tools and our method are ML-based, and used the same training and validation data in every case.

## C  Additional Experiments

**1) Architecture exploration:**  We explored the alternative architectures listed in Table 5, which shows the performance of different network architectures with different numbers of layers with and without an attention mechanism. The reported metrics are averages of the five-fold cross-validation results obtained with the *human* dataset.

Table 5: Performance of different types of neural network, assessed in terms of five-fold cross-validation results from the *human* dataset (the same table as Table 4 in the main text; repeated here for the sake of convenience). The number of stacked layers is shown in brackets. ATT means that an attention mechanism was included, and a BiLSTM is a bi-directional LSTM. The configuration that we finally adopted is shown in row 6.

| No. | Type | SE | SP | F-score | g-mean |
|-----|------|-----|-----|---------|--------|
| 1 | 1D-CNN(2) | 0.745 | 0.978 | 0.771 | 0.853 |
| 2 | 1D-CNN(2)+LSTM(2) | 0.707 | 0.976 | 0.738 | 0.830 |
| 3 | 1D-CNN(2)+LSTM(2)+ ATT | 0.691 | 0.979 | 0.739 | 0.822 |
| 4 | LSTM(2) | 0.666 | **0.988** | 0.751 | 0.810 |
| 5 | LSTM(1) + ATT | 0.781 | 0.987 | 0.824 | 0.878 |
| 6 | LSTM(2) + ATT (proposed) | **0.799** | **0.988** | **0.839** | **0.888** |
| 7 | BiLSTM(1) + ATT | 0.783 | 0.987 | 0.827 | 0.879 |
| 8 | BiLSTM(2) + ATT | 0.795 | 0.987 | 0.834 | 0.886 |

Table 6: Performance evaluation with varying numbers of LSTM units.

| 1st LSTM layer | 2nd LSTM layer | SE | SP | F-score | g-mean |
|----------------|----------------|-----|-----|---------|--------|
| 20 | 10 | 0.799 | **0.988** | **0.839** | **0.888** |
| 20 | 20 | **0.802** | 0.985 | 0.829 | **0.888** |
| 40 | 10 | 0.785 | **0.988** | 0.829 | 0.880 |
| 40 | 20 | 0.785 | 0.987 | 0.827 | 0.880 |
| 40 | 40 | 0.773 | 0.987 | 0.820 | 0.873 |
| 100 | 10 | 0.783 | 0.985 | 0.819 | 0.878 |
| 100 | 20 | 0.770 | 0.985 | 0.812 | 0.871 |
| 100 | 40 | 0.750 | 0.985 | 0.798 | 0.859 |
| 100 | 100 | 0.767 | 0.984 | 0.804 | 0.868 |

In rows 1–3 of the table, we show the results for CNNs with or without LSTM networks. There are one-dimensional CNNs with a layer of 32 filters and another layer of 64 filters. Both filter and pooling sizes were set to three. CNNs are good at discovering short motifs, and the simple 1D-CNN in row one showed comparable performance to our scheme; but the addition of LSTM networks (rows 2 and 3) somewhat degraded its performance. This can be explained because extracting short motifs discards the long-term sequential information inherent in the sequence of nucleotides and structure elements. Rows 4–6 of the table show results from LSTM networks. The two-layer LSTM network without an attention mechanism (row 4) showed the lowest F-score of 0.751. The performance of these networks can be greatly improved by an attention mechanism (rows 5 and 6). As stated in Section 4.3, this result suggests that the long-term dependency between the front and rear part of a sequence (captured by an attention mechanism) is essential in pre-miRNA identification. Stacking two LSTM layers (row 6) produced better performance than a single LSTM layer (row 5). Stacking further layers caused underfitting due to the limited data, and we omitted the results. Rows 7–8 show results obtained with bi-directional LSTM (BiLSTM) networks. We could not find noticeable improvement, even though a BiLSTM layer has the twice parameters of an LSTM layer, and it means that considering attentive sequential information in one direction is sufficient for the current task.

**2) LSTM hyperparameter tuning:** To find the optimal number of units in each of the two LSTM layers in our model, we measured its performance with the configurations shown in Table 6. Using 20 units for the first LSTM layer and 10 units for the second LSTM layer produced the best result. Using more units in either layer degraded the performance, probably due to overfitting.

Table 7: Performance evaluation with soft and hard encoding.

| | SE | SP | F-score | g-mean |
|--|-----|-----|---------|--------|
| hard encoding (one-hot) | 0.799 | **0.988** | **0.839** | 0.888 |
| soft encoding | **0.821** | 0.982 | 0.830 | **0.898** |

**3) Soft vs. hard encoding:**    We use one-hot encoding (e.g., [0 1]) for representing sequences and structures in our method. An alternative would be to learn a soft version of the one-hot encoding by training (e.g., [0.2 0.8]). Such an approach has been effective in recent studies on natural language processing [9]. In our experiments, however, a soft encoding did not give significant advantages, as shown in Table 7, despite additional training time.

**4) Running time:**    The total training time of the proposed method for the *human* dataset was approximately 1 hours (12 seconds $\times$ 300 epochs) per fold. Triplet-SVM required the shortest training time of 2.7 seconds (1 second for testing). The second shortest training time was 13 minutes (2 seconds for testing) for miRBoost. The training of CSHMM and microPred was relatively time-consuming, taking 10 hours and 30 hours respectively. Deep learning-based methods are generally considered to be slower than feature-based tools. However, testing with our method was as fast as the fastest alternatives (less than 2 seconds). MIReNA (8.3 seconds for testing) is a rule-based method which does not require any training, and it was excluded from this comparison.