[Reviews · NeurIPS 2017]

Reviewer 1



In this paper, the authors address the prediction of miRNA precursors from both RNA sequence and secondary structure with a recurrent neural network. More precisely, the secondary structure is predicted using RNAfold. Hence each position in the sequence is annotated with two characters, one among {A,T,C,G} to encode the sequence itself and the other among {(,:,)} corresponding to the secondary structure according to classical dot-bracket notation. These are transformed into a 16-dimensional vector using one-hot encoding. Now that the problem is about classifying sequences, it makes sense to use state-of-the-art units for sequence classification, that is to say, LSTMs with an attention mechanism. Attention weights then show which parts of the sequences are activated in each of the positive and negative classes. The core contribution of the paper hence build quite naturally on the literature, but to the best of my knowledge this had never been done before and this appears to be well done. I have a few concerns nevertheless, namely: - I am not very familiar with LSTMs, and I have a hard time understanding the architecture of the network, in particular, relating Figure 2 with the equations in Section 3.2. - In the Introduction, the authors mention two neural-network-based algorithms for the same task, namely MiRANN and DP-miRNA. Why aren't those evaluated in the experiments? - Regarding Table 2, are those AUCs cross-validated or reported on the test set? It seems strange that the proposed method outperforms the others in terms of AUC on 'human' but not in terms of any other metric. - At the end of Section 4.1, the authors report p-values on the significance of their results. I would rather like to see whether their method significantly outperforms the closest one (for instance, is the AUC of 0.9850 on human significantly better than that of 0.9802?)

Reviewer 2



The paper presents an LSTM model with an attention mechanism for classifying whether an RNA molecule is a pre-microRNA from its sequence and secondary structure. Class weights are incorporated into log-loss to account for class imbalance in the datasets used. The proposed method is extensively evaluated against 5 other existing methods on 3 datasets, and is shown to outperform the existing methods in most cases. The paper then attempts to give some insight into the features that are important for achieving good performance. First, by showing that secondary structures are largely responsible, but sequence features give a small boost, and second, by interpreting the attention weights using an adapted version of class activation mapping (proposed in an earlier CVPR paper). Results using different architectures and hyperparameters are also shown. The paper presents an interesting application of LSTMs to biological data, and has some novel elements for model interpretation. The main issue I have with this paper is in how the method is evaluated. If the goal is to find new pre-microRNAs, especially those that have noncanonical structures (as stated in the abstract) special care must be taken in the construction of the training and test sets to make sure the reported performance will be reflective of performance on this discovery task. This is why in ref [10] describing miRBoost, methods are evaluated based on ability to predict completely new pre-microRNAs that were not previously found in previous versions of miRBase (included as 'test' dataset in this paper). While holding out 10% of the dataset as a test set may seem to also be sufficient, it results in a far smaller evaluation set of positives (1/5 - 1/10) than using the 'test' dataset, as well as possibly allowing the models to learn structural features of newly discovered RNAs, some of which may have been randomly placed in the training set. An even stricter evaluation would use "structurally nonhomologous training and test data sets", as proposed in Rivas et al. (PMID:22194308), that discusses the importance of doing so for properly evaluating methods on the related RNA secondary structure prediction task. If a model is able to perform well on this stricter evaluation, where test RNA sequences are structurally different from training sequences, one can then be more confident in the ability to discover structurally novel pre-microRNAs. A related comment on the evaluation is that the AUROC metric is unsuitable for tasks where only performance on the positive class is important (as in this case) and the area under the precision-recall curve (AUPR), or even better, the area under the precision-recall-gain curve (AUPRG; Flach and Kull NIPS 2015) should be used. The adaptation of the class activation mapping method to the LSTM model is interesting and the paper hints that it could indeed be useful in uncovering new biological insight. It would strengthen the section if features common to the negatively predicted examples could be discussed as well. The attention mechanism seemed to make a big difference to the performance - it would be interesting if it was possible to dissect how/why this is occurring in the model (and on this task). Does the attention mechanism make a bigger difference for longer sequences? And is it possible to show some of the long range dependencies that the attention mechanism picks up? Other questions for the authors: - In section 4.3 were the chosen examples actually correctly classified as positives and negatives respectively? - Line 175 - how were the dropout rates selected empirically? Using cross validation? - In the discussion it is mentioned that additional data modes can be added as separate network branches. Why was this approach not taken for the structure data and instead an explicit encoding that simultaneously considers structure and sequence was used? Typos: Line 91: "long-term dependency" -> "long-term dependencies" Table 1: "Positive exmaples" -> "Positive examples" Table 2 caption: "under curve of receiver operation characteristic" -> "area under receiver operating characteristic curve" Table 3: microPred (test), cross-species, 0.985 should be highlighted instead Figure 4 caption: "RNN ouputs" -> "RNN outputs" Line 237: "technqiue" -> "technique" Reference 10 should be Tran et al. not Tempel et al.

Reviewer 3



The authors present a nice paper on the first deep learning attempt at pre-miRNA identification, and demonstrate results (marginally) better than state of the art approaches. The main strength of the paper is the originality of use of RNN for this task (given that this is the first such attempt), and the fact that they are able to demonstrate even subtle improvements over methods that employ hand-crafted features, indicating that their approach effectively extracts features in a manner that is superior to the effectiveness of human experts with domain expertise, a compelling demonstration. Some positive attributes: The paper is well written and clear, I liked the analysis of seq vs structure vs both (demonstrating that seq doesn't add much!), and I am insufficiently familiar with RNA prediction to speak to the novelty of this, but I liked their method for encoding seq+structure as a matrix of one hot vectors. Additional comments: -Section 4.3, in which an effort is made to extract the useful features, is pertinent and interesting, but a bit poorly developed. What insights were gained from this analysis? It seemed only to confirm what the researchers would have guessed a priori, validating that the method works, but not providing any useful insights. -The main weakness of the paper is that it appears as an assembling of components in order to try out RNN on this prediction task, precisely the opposite of what the authors claim at the bottom of section 1. In what way was this study a careful crafting of a pipeline optimized to this task (as claimed), rather than a somewhat arbitrary compilation of layers? Similarly, a number of settings are presented with neither reference nor justification (e.g. last line of section 3.2. Why are those activation functions used for their respective layers?). All together, this results in a possibility that the approach is overfit, not the model itself (as nicely demonstrated, the final model does NOT overfit these data) but the arbitrary-seeming nature of the path towards the final model makes it seem like the process itself overfits and 'cherry picks' this model. The authors could address this weakness by clearly stating their contributions in this study (how each component of the pipeline was chosen, how it fits this specific task), by explaining why they tried the configurations listed in Table 4, and by explaining their 'design decisions' a bit more thoroughly. Table 4 summarizes their architecture exploration on one of the datasets. It was unclear to me if they used only that one dataset to select among the architectures? Or if they are only presenting one set of data in the table, but all 3 were used in selecting the final model architecture? They should clarify this point. If in fact all 3 datasets were used, if it is possible to leave one out for this stage and re-select the architecture, then check its ability to generalize to the final dataset, this would go far to assuage concerns of the process leading to an overfit model.